# Learning Visual Representations for Transfer Learning by Suppressing Texture

## Abstract

Recent literature has shown that features obtained from supervised training of CNNs may over-emphasize texture rather than encoding high-level information. In self-supervised learning in particular, texture as a low-level cue may provide shortcuts that prevent the network from learning higher level representations. To address these problems we propose to use classic methods based on anisotropic diffusion to augment training using images with suppressed texture. This simple method helps retain important edge information and suppress texture at the same time. We empirically show that our method achieves state-of-the-art results on object detection and image classification with eight diverse datasets in either supervised or self-supervised learning tasks such as MoCoV2 and Jigsaw. Our method is particularly effective for transfer learning tasks and we observed improved performance on five standard transfer learning datasets. The large improvements (up to 11.49%) on the Sketch-ImageNet dataset, Synthetic-DTD dataset and additional visual analyses with saliency maps suggest that our approach helps in learning better representations that better transfer.

## 1 Introduction

Deep convolutional neural networks (CNNs) can learn powerful visual features that have resulted in significant improvements on many computer vision tasks such as semantic segmentation (Shelhamer et al., 2017), object recognition (Krizhevsky et al., 2012), and object detection (Ren et al., 2015). However, CNNs often fail to generalize well across datasets under domain-shift due to varied lighting, sensor resolution, spectral-response etc. One of the reasons for this poor generalization is CNNs' over reliance on low-level cues like texture (Geirhos et al., 2018). These low-level cues and texture biases have been identified as grave challenges to various learning paradigms ranging from supervised learning (Brendel & Bethge, 2019; Geirhos et al., 2018; Ringer et al., 2019) to self-supervised learning (SSL) (Noroozi & Favaro, 2016; Noroozi et al., 2018; Doersch et al., 2015; Caron et al., 2018; Devlin et al., 2019).

We focus on learning visual representation that are robust to changes in low-level information, like texture cues. Specifically, we propose to use classical tools to suppress texture in images, as a form of data augmentation, to encourage deep neural networks to focus more on learning representations that are less dependent on textural cues. We use the Perona-Malik non-linear diffusion method (Perona & Malik, 1990), robust Anistropic diffusion (Black et al., 1998), and Bilateral filtering (Tomasi & Manduchi, 1998) to augment our training data. These methods suppress texture while retaining structure, by preserving boundaries.

Our work is inspired by the observations that ImageNet pre-trained models fail to generalize well across datasets (Geirhos et al., 2018; Recht et al., 2019), due to over-reliance on texture and low-level features. Stylized-ImageNet (Geirhos et al., 2018) attempted to modify the texture from images by using style-transfer to render images in the style of randomly selected paintings from the Kaggle paintings dataset. However, this approach offers little control over exactly which cues are removed from the image. The resulting images sometimes retain texture and distort the original shape. In our approach (Fig. 1), we suppress the texture instead of modifying it. We empirically show that this helps in learning better higher level representations and works better than CNN-based stylized augmentation. We compare our approach with Gaussian blur augmentation, recently used in (Chen

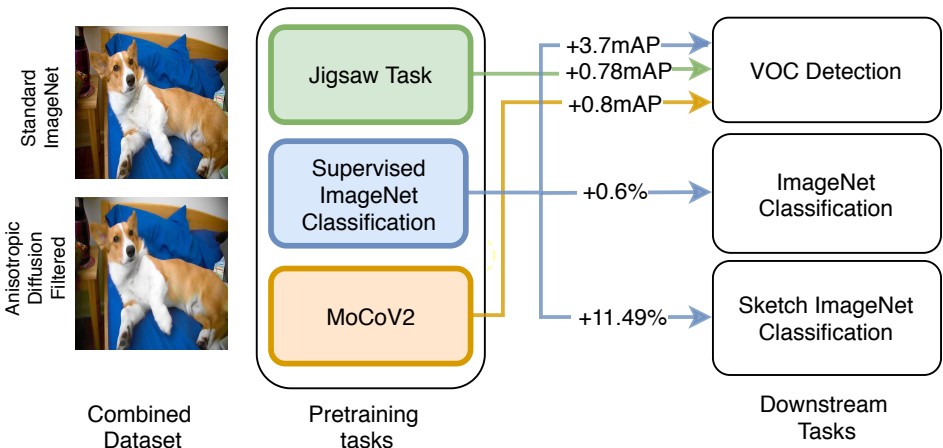

Figure 1: An overview of our approach. We propose to augment the ImageNet dataset by a version of the dataset with Anisotropic diffused images. The use of this augmentation helps the network rely less on texture information and increases performance in diverse experiments.

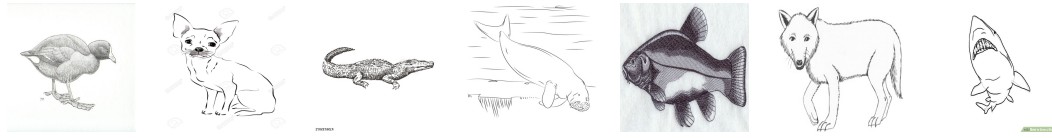

Figure 2: Examples of images from Sketch-ImageNet. Images have very little or no texture, which implies texture will have little to no impact on object classification.

et al., 2020a;b), and show that Anisotropic-filtering for texture suppression is better, because Gaussian blur can potentially suppress edges and other higher-level semantic information as well.

Our approach yields consistent improvements in both supervised and self-supervised learning settings for learning representations that generalize well across different datasets. For the supervised setting, we pre-train on ImageNet, and test on eight different datasets including ImageNet, Pascal VOC (Everingham et al., 2009), Synthetic-DTD (Newell & Deng, 2020), CIFAR 100 (Hendrycks et al., 2019), Sketch ImageNet (Wang et al., 2019), etc. For self-supervised setting, we used two learning frameworks: Jigsaw (Noroozi & Favaro, 2016), and MoCoV2 (Chen et al., 2020b). MoCo (He et al., 2019) and MoCoV2 (Chen et al., 2020b) have achieved competitive performance on ImageNet classification and have outperformed supervised pre-trained counterparts on detection and segmentation tasks on the PASCAL VOC (Everingham et al., 2009) and COCO datasets (Lin et al., 2014). Our texture-suppressing augmentation consistently outperforms MoCoV2, which uses Gaussian blurring, and Jigsaw on transfer learning experiments on VOC classification, detection, segmentation benchmarks, and also on classification tasks for other transfer learning datasets, including DTD (Cimpoi et al., 2014), Cars (Krause et al., 2013), Aircraft (Maji et al., 2013), etc.

Overall, we achieve significant improvements on several benchmarks:

- In a set of **eight** diverse datasets, our method exhibits substantial improvements (as high as +11.49% on Sketch ImageNet and 10.41% on the Synthetic-DTD dataset) in learning visual representations across domains.

- We also get improvements in same domain visual recognition tasks on ImageNet validation (+0.7%) and a label corruption task (Hendrycks et al., 2019).

- We achieve state-of-the-art results in self-supervised learning on VOC detection and other transfer learning tasks.

## 2 RELATED WORK

In this section, we review relevant methods that aim to remove texture cues from images to reduce the dependency of CNNs on low-level cues. Since we also experiment with the application of our method in self-supervised learning, we review recent work in this area as well.

**Reliance on Low-Level Texture Cues.** Recent studies have highlighted that deep CNNs can leverage low-level texture information for classification on the ImageNet dataset. Contrary to the popular belief that CNNs capture shape information of objects using hierarchical representations (LeCun et al., 2015), the work in (Geirhos et al., 2018) revealed that CNNs trained on ImageNet are more biased towards texture than shape information. This dependency on texture not only affects generalization, but it can also limit the performance of CNNs on energing real-world use-cases, like few-shot image classification (Ringer et al., 2019). Brendel & Bethge (2019) showed that a bag of CNNs with limited receptive field in the original image can *still* lead to excellent image classification performance. Intuitively, a small receptive field forces the CNNs to heavily rely on local cues vs. learning hierarchical shape representations. This evidence strongly suggests that texture alone can yield competitive performance on ImageNet and the fact that it's relatively easier to learn vs. hierarchical features may explain deep CNNs' bias towards texture. In order to reduce reliance on texture, Stylized-ImageNet (Geirhos et al., 2018) modified the ImageNet images into different styles taken from the Kaggle Painter by Numbers dataset. While trying to remove texture, this approach could also significantly affect the shape. Also, there isn't an explicit control over the amount of removed texture. Moreover, this method may not be directly applicable to self-supervised learning because the fixed number of possible texture patterns result in images with strong low-level visual cues resulting in shortcuts. We show that the accuracy on downstream tasks, when MoCoV2 and Jigsaw are trained with Stylized-ImageNet, decreases dramatically (Table 1 Supplementary). We, on the other hand, use Perona-Malik's anisotropic diffusion (Perona & Malik, 1990) and bilateral filtering (Tomasi & Manduchi, 1998) as ways of suppressing texture in images. These methods remove texture without degrading the edge information. Consequently, the shape information of the objects are better preserved. Also, these methods provide finer control over the level of texture suppression. Suppressing the texture in training images forces the CNN to build representations that put less emphasis on texture. We show that such data augmentation can lead to performance improvements in both supervised and self-supervised settings. We also distinguish our work from other data augmentation strategies like Auto-Augment (Cubuk et al., 2018) which uses Reinforcement Learning to automatically search for improved data augmentation policies and introduces Patch Gaussian Augmentation, which allows the network to interpolate between robustness and accuracy (Lopes et al., 2019). The motivation behind our proposed approach is to suppress the reliance of CNNs on low-level cues and encourage CNNs to learn representations that are less dependent on texture. One of the most recent works in this area which has been brought to our attention during the reviewer discussion phase is Informative-dropout (Shi et al., 2020). Informative-dropout (Shi et al., 2020) tries to reduce texture by using dropout based method which zeros out neurons by a high probability if the input patch contains less self-information. We however use classical methods like Anistropic diffusion (Perona & Malik, 1990) to suppress texture.

**Self-Supervised Learning.** To demonstrate the importance of removing texture in the self-supervised setting, we consider two pretext tasks. The first pretext task is Jigsaw (Noroozi & Favaro, 2016) which is a patch based self-supervised learning method that falls under the umbrella of visual permutation learning (Cruz et al., 2017; 2018) . Some of the most recent self-supervised methods are contrastive learning based methods (He et al., 2019; Caron et al., 2018; Hénaff et al., 2019; Hjelm et al., 2018; Misra & van der Maaten, 2019; Chen et al., 2020a;b). In Caron et al. (2018), the authors have proposed using contrastive losses on patches, where they learn representations by predicting representations of one patch from another. In MoCo (He et al., 2019), a dynamic dictionary is built as a queue along with a moving average encoder. Every image will be used as a positive sample for a query based on a jittered version of the image. The queue will contain a batch of negative samples for the contrastive losses. MoCo has two encoder networks. The momentum encoder has weights updated through backpropagation on the contrastive loss and a momentum update. In MoCoV2, Gaussian blur and linear projection layers were added that further improve the representations. MoCo and MoCoV2 have shown competitive results on ImageNet classification and have

outperformed supervised pre-trained counterparts on seven detection/segmentation tasks, including PASCAL VOC (Everingham et al., 2009) and COCO (Lin et al., 2014).

**Transfer Learning.**   Transfer learning is one of the most important problems in computer vision due to difficulty in collecting large datasets across all domains. In this work, we discuss transfer learning in context of ImageNet. A lot of early datasets were shown to be too small to generalize well to other datasets (Torralba & Efros, 2011). Following this, many new large-scale datasets were released (Deng et al., 2009; Lin et al., 2014), which are believed to transfer better. However, recent results have shown that these datasets do not generalize well in all cases (Recht et al., 2019; Kornblith et al., 2019). Kornblith et al. (2019) showed that ImageNet features generally transfer well, but do not transfer well to fine-grained tasks. We show results of transfer learning on some the datasets that were used by Kornblith et al. (2019).

## 3    METHODS

Texture and other visual cues may bias CNNs towards over-fitting on these cues. This may lead to brittleness when these cues change in new domains. CNN-based classifiers have been shown to exploit textures rather than shapes for classification (Geirhos et al., 2018; Brendel & Bethge, 2019). We aim to reduce the prominence of texture in images, and thus encourage networks trained on them to learn representations that capture better higher level representations.

**Gaussian Blur.**  Gaussian blurring is the most popular smoothing methods in computer vision, and it has been recently proposed as data augmentation for SSL (Chen et al., 2020a;b). However, along with low-level texture, Gaussian filtering also blurs across boundaries, diminishing edges and structural information.

**Anisotropic Diffusion.** We propose to use Anisotropic Diffusion Filters or ADF (Perona & Malik, 1990), which keep the shape information coherent and only alter low-level texture. We specifically use Perona-Malik diffusion (Perona & Malik, 1990). These filters smooth the texture without degrading the edges and boundaries. Intuitively, it will encourage the network to extract high-level semantic features from the input patches. Interestingly, we find that a relatively modest amount of smoothing suffices to reduce texture shortcuts.

Perona-Malik diffusion smooths the image using the differential diffusion equation:

$$\frac{\partial I}{\partial t} = c(x, y, t)\Delta I + \nabla c \cdot \nabla I \tag{1}$$

$$c(x, y, t) = e^{-(\|\nabla I(x,y,t)\|/K)^2} \tag{2}$$

where $I$ is the image, $t$ is the time of evolution, $\nabla$ is the Laplacian operator, and $(x, y)$ is a location in the image. The amount of smoothing is modulated by the magnitude of the gradient in the image, through $c$. The larger the gradient, the smaller the smoothing at that location. Therefore, after applying Anisotropic diffusion we obtain images with blurred regions but edges are still prominent. Fig. 3 shows some examples of the application of the filter. Since the ADF reduces the texture in the image without replacing it, the domain gap between images is not large, while in the case of Stylized ImageNet, the domain shift will be large. We also experiment with a few other texture removing methods like robust Anistropic diffusion (Black et al., 1998) and Bilateral filtering (Tomasi & Manduchi, 1998). However, empirically we find that the most simple Anistropic diffusion method has the best results as discussed in Section 4.2. Recently, there has been some work on removing textures using deep learning as well (Xu et al., 2014; Liu et al., 2016; Lu et al., 2018). We find, though, that fast and simple classical methods work well on our tasks.

For all our experiments we create a dataset 'Anisotropic ImageNet' by combining ADF filtered ImageNet images with standard ImageNet.

## 4    EXPERIMENTS

We start by briefly describing the datasets used in our experiments. We then show that ADF is particularly effective when there is domain shift, supporting our hypothesis that variation in texture is a significant effect of domain shift. We show this in both SSL and supervised settings. The effect

| Normal ImageNet | Anisotropic Diffusion Filtered | Cartoon Images | Gaussian Images | Bilateral Images |
|---|---|---|---|---|

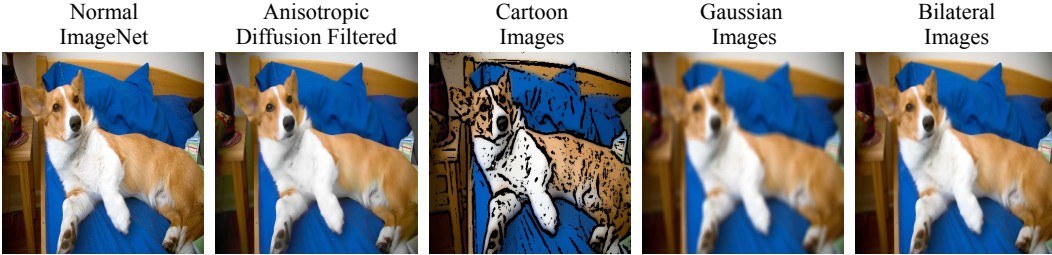

Figure 3: Four different methods for reducing texture in images.

is larger when we transfer from ImageNet to datasets such as Sketch Imagenet (Wang et al., 2019) and Synthetic-DTD (Newell & Deng, 2020), where the domain shift is larger. We also show that when there is no domain shift, our method is competitive with other methods.

**Datasets.** In all experiments, we use ImageNet training set as the source of our training data. For object detection and semantic segmentation, we evaluate on Pascal VOC 2007 and VOC 2012. For label corruption, we evaluate on CIFAR100. For the downstream task is classification we evaluate on DTD (Cimpoi et al., 2014), Sketch-ImageNet (Wang et al., 2019), Birds(Wah et al., 2011), Aircraft(Maji et al., 2013), Stanford Dogs(Khosla et al., 2012), Stanford Cars (Krause et al., 2013), and the ImageNet validation dataset.

**Experimental Details.** For SSL we build on MoCoV2 (Chen et al., 2020b). For supervised learning, we use the ResNet50 (He et al., 2015) model, closely following (Geirhos et al., 2018). After training on Anisotropic ImageNet, we fine-tune our model on the standard ImageNet training set following the procedure of (Geirhos et al., 2018).

**Hyper-parameters for Anisotropic Diffusion.** We set the conduction coefficient ($K$) of Anisotropic Diffusion to 20 and a total of 20 iterations are used. We use the MedPy implementation. All other hyper-parameters are described in supplementary material.

### 4.1 TRANSFER LEARNING FOR SELF-SUPERVISED LEARNING

We first experiment with Anisotropic ImageNet on Self-Supervised methods. We have double the number of images (Anisotropic images + normal images) as compared to normal ImageNet. So for fair comparison, we only train our methods for half the number of epochs as compared to training with just ImageNet. We then fine-tune the network pre-trained on the Anisotropic ImageNet for the downstream tasks including image classification, object detection, and semantic segmentation on PASCAL VOC, and other transfer learning datasets. Since, we are removing low-level cues from the images, we expect to see better results when we are transferring to different datasets.

**MoCo V2.** We evaluate our method with MoCo V2 (Chen et al., 2020b), which is the state-of-the-art in SSL. The original MoCoV2 used Gaussian blurring with 0.5 probability as data augmentation. In our case, we add Anisotropic diffusion on the images with 0.5 probability, and for the remaining 50% of the images we apply Gaussian blurring with 0.5 probability. So, in our setup every image has 0.5 probability of coming from Anisotropic ImageNet, 0.25 of Gaussian blurring, and 0.25 of being normal ImageNet. Also, the number of iterations on Anisotropic filtering is chosen randomly between 10 to 20. We conduct two sets of experiments on MoCoV2 for object detection. In the first setup, starting from a MoCoV2 initialization, we train a Faster R-CNN (Ren et al., 2015) with C4-backbone, which is fine-tuned end-to-end. In the second setup, we again initialize Faster-RCNN from the MoCoV2-trained network, and only train the region proposal, classification, and box-regression layers, and keep the rest of the layers unchanged (the performance for this case is marked as AP*$_{50}$ in Table 1). In both setups, training is done on VOC(07+12) trainval set and we evaluate on the VOC07 test.

For both setups, we achieve the state-of-the-art performance for object detection on VOC Dataset. In the first setup, we show improvements on COCO-based evaluation metrics (i.e., AP$_{50}$, AP$_{0.05:0.05:0.95}$, AP$_{75}$) as shown in first three columns of Table 1, and achieve new state-of-the-art performance on object detection. In the second setup, Table 1 shows that the baseline detection accuracy for MoCoV2 trained on ImageNet is 66.5 mAP, and the one trained with our method is 67.3 mAP. We also observe improvement of 0.5 mean IoU on semantic segmentation (Long et al.,

Table 1: Comparison with the state of the art methods in SSL. We note that using Anisotropic diffusion with MoCoV2 improves performance on VOC detection and Semantic Segmentation (SS). We test on two types of metrics for object detection: first is COCO-based metrics as used in Chen et al. (2020b) and the second metric AP*$_{50}$ uses frozen backbone Ren et al. (2015). We achieve the state-of-the-art results on all metrics. We also improve performance over the baseline on the semantic segmentation (SS) task (Long et al., 2015). We also compare with supervised ImageNet pretrained backbone as well. MoCoV2 has been trained on 200 epochs.

| Methods | AP$_{50}$ | AP$_{0.50:0.05:0.95}$ | AP$_{75}$ | AP*$_{50}$ | mIoU (SS) |
|---|---|---|---|---|---|
| Stylized ImageNet (Geirhos et al., 2018) | 43.5 | 28.80 | 33.7 | - | - |
| Supervised ImageNet | 81.3 | 53.5 | 58.8 | 70.1 | 53.5 |
| MoCo V2 (Chen et al., 2020b) | 82.4 | 57.0 | 63.6 | 66.5 | 55.5 |
| MoCo V2 Anistropic (Ours) | **82.8** | **57.4** | **64.2** | **67.3** | **56.1** |

Table 2: Transfer learning across different datasets. Note that our approach leads to improvements in both supervised and self-supervised learning set-up.

| Dataset | Aircraft | Birds | Dogs | Cars | DTD |
|---|---|---|---|---|---|
| Supervised (Reproduced) | 90.88 | 90.3 | 85.35 | 92.1 | 72.66 |
| Supervised Anistropic (Ours) | **91.67** | **91.42** | **86.40** | **93.1** | **73.03** |
| SimCLR (Chen et al., 2020a) | 88.1 | – | – | 92.1 | 73.2 |
| MoCo V2 (Chen et al., 2020b)(Reproduced) | 91.57 | 92.13 | 87.13 | 92.8 | 74.73 |
| MoCo V2 Anistropic (Ours) | **92.05** | **92.76** | **87.92** | **93.5** | **75.12** |

2015) over MoCo V2 baseline. These results show that in case of transfer learning, we improve across different datasets. More details can be found in the supplementary material. Our method is not bound to a particular pretext task and in the supplementary material we show that our method leads to improvements with the Jigsaw (Noroozi et al., 2018) task as well.

These results suggest that training the network on the Anisotropic ImageNet dataset forces it to learn better representations. This is consistent with our hypothesis that Anisotropic diffusion leads to smoothing of texture in images. This forces the network to be less reliant on lower-level information to solve the pretext task and, hence, learn representations that focus on higher-level concepts.

**Experiments with Stylized ImageNet on MoCoV2 and Jigsaw.** We now show experiments that indicate that, while effective in a supervised setting, Stylized ImageNet does not help with SSL. We train a model with MoCoV2 and Jigsaw as pretext tasks on the Stylized-ImageNet (SIN) dataset (Geirhos et al., 2018) and fine-tune on the downstream tasks of object detection and image classification on PASCAL VOC. In Table 2 (and Table 2 in supplementary), we show that there is a huge drop in performance. One reason for this failure using the SIN dataset could be that the model is able to memorize the textures in the stylized images since it only has 79,434 styles. This is not a problem in the original fully-supervised setting where the authors used SIN for supervised image classification. In that case, the network can learn to ignore texture to discriminate between classes.

## 4.2 TRANSFER LEARNING FOR SUPERVISED LEARNING

As shown in the last section, suppressing texture leads to performance improvements in the case of domain transfer with SSL. In this section, we also show improvements on supervised learning and domain transfer.

### 4.2.1 ACROSS DOMAINS

We hypothesize that learning the texture bias is the most harmful when it comes to domain transfer. Thus, we first design the challenging experimental setup for learning visual representation; learning it for different domains.

Table 3: Experiments with Sketch-ImageNet. Use of Anisotropic ImageNet shows that our method is better at capturing representation that are less dependent on texture.

| Method | Top-1 Accuracy | Top-5 Accuracy |
|---|---|---|
| ImageNet Baseline | 13.00 | 26.24 |
| Stylized Baseline | 16.36 | 31.56 |
| Anisotropic (Ours) | **24.49** | **41.81** |

Table 4: Comparison using different texture removing methods, with different hyper-parameters for Anisotropic diffusion methods. We observe that the most simple (Perona & Malik, 1990) performs the best and removing more texture from images does not improve performance.

| Method | # Iterations | Top-1 Acc | Top-5 Acc | Obj. Det. |
|---|---|---|---|---|
| Baseline Supervised | - | 76.13 | 92.98 | 70.7 |
| Perona Malik (Perona & Malik, 1990) | 20 | **76.71** | **93.26** | **74.37** |
| Perona Malik (Perona & Malik, 1990) | 50 | 76.32 | 92.96 | 73.80 |
| Robust AD (Black et al., 1998) | 20 | 76.58 | 92.96 | 73.33 |
| Robust AD (Black et al., 1998) | 50 | 76.64 | 93.09 | 73.57 |
| Gaussian Blur | - | 76.21 | 92.64 | 73.26 |
| Cartoon ImageNet | - | 76.22 | 93.12 | 72.31 |
| Bilateral ImageNet | - | 75.99 | 92.90 | 71.34 |

**Sketch-ImageNet.** For a cross-domain supervised learning setup, we chose to use the Sketch-ImageNet dataset. Sketch-ImageNet contains sketches collected by making Google image queries "sketch of X", where "X" is chosen from the standard class names of ImageNet. The sketches have very little to no texture, so performance on Sketch-ImageNet is a strong indicator of how well the model can perform when much less texture is present. Sketch-ImageNet has been collected in the same fashion as Recht et al. (2019), implying that validation set is different compared to the original ImageNet validation set. As shown in Table 3, the difference between the Anisotropic model and the baseline model is 11.49% for Top-1 accuracy, This result implies that our model captures representations that are less dependent on texture as compared to standard ImageNet and Stylized ImageNet.

**Synthetic-DTD Dataset.** To better demonstrate the effectiveness of less texture dependent representations, we used the dataset introduced by (Newell & Deng, 2020). This dataset provides four variations in images: Texture, color, lighting, and viewpoint. It contains 480,000 training Images and 72,000 testing Images. In this dataset, we made sure that texture information during training and testing are completely different. So, the texture is not a cue when we use this dataset. We evaluated our Anisotropic model on this dataset and compared against the baseline normal ImageNet model. The Anisotropic model achieves a performance boost of 10.41% in classification which suggests that we are indeed able to learn texture agnostic feature representations.

**Other Datasets - Aircraft, Birds, Dogs, DTD and Cars.** We further evaluate our method on image classification task on five different fine-grained classification datasets. We observe improvement on image classification across five datasets in Table 2. These results suggest that in case of domain shift, higher level semantics are more important and capturing them helps in better transfer learning performance.

**Object Detection.** The biggest improvement we observe on transfer learning is on object detection on Faster-RCNN (Ren et al., 2015) as shown in Table 4. This improvement on object detection suggests that we are able to capture more high-level semantics which helps us in transfer learning performance on object detection.

### 4.2.2 SAME DOMAIN

We also observe consistent performance improvements in the same domain setups.

**ImageNet.** In Table 4, we show results by using Anisotropic ImageNet for supervised classification. We observe that Anisotropic ImageNet improves performance in both ImageNet classification and

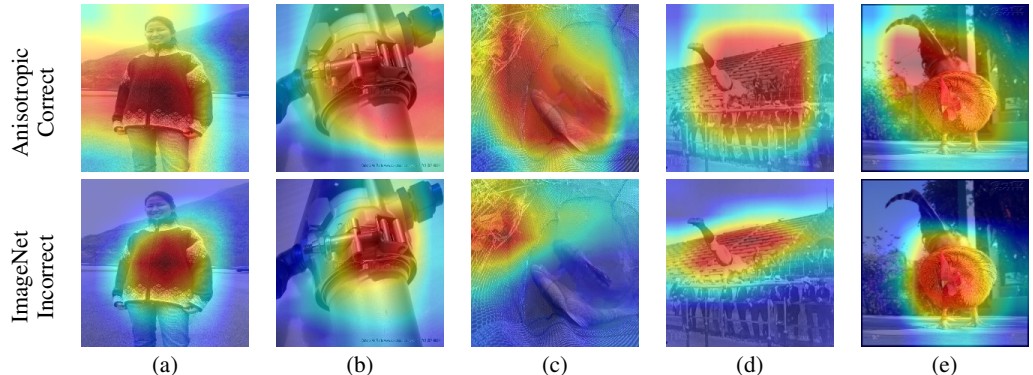

Figure 4: Saliency maps using GradCam. The text on the left of the row indicates whether Anisotropic model or ImageNet model was used. The figure shows the saliency maps where Anisotropic model gave correct predictions and ImageNet model gave wrong predictions. The failure of ImageNet model might be due to it not attending to whole object.

object detection. For Gaussian blurring experiments, we closely follow Chen et al. (2020b) and add a Gaussian blur operator with random radius from 10 to 20 and train in a similar manner to Stylized ImageNet (Geirhos et al., 2018). So, this result shows that Anisotropic ImageNet is similar to Stylized ImageNet and is a better alternative to Gaussian blurring. We also observe that Gaussian blurring does not perform as well as Anisotropic ImageNet in terms of ImageNet top-1 and VOC object detection performance. Hence, blurring the image completely without respecting boundaries and edges does not improve performance as much as Anisotropic diffusion.

**Different Texture Removing Methods.** We also provide results using different texture removing methods and different hyper-parameters for Anistropic diffusion in Table 4. We observe that as we increase the number of iterations and remove more and more texture from images, performance starts to degrade possibly due to the difference that comes in the data distribution after removing texture information. The most simple texture removing method (Perona & Malik, 1990) has the best results. We also show results on the task of label corruption in supplementary material.

### 4.3 VISUAL ANALYSIS BY SALIENCY MAPS

We now visually analyze the results by the saliency maps, which are produced by different networks. We use GradCam (Selvaraju et al., 2016) to calculate the saliency maps. In Fig 4, we show the saliency maps produced by networks trained using the combined dataset and the original ImageNet dataset. We observe that Anisotropic ImageNet has saliency maps that spread out over a bigger area and that include the outlines of the objects. This suggests that it attends less to texture and more to overall holistic shape. In contrast, ImageNet trained models have narrower saliency maps that miss the overall shape and focus on localized regions, suggesting attention to texture.

In Fig. 4(a-e), we present the examples where the Anisotropic model gives the correct prediction, and the ImageNet model fails. For example in Fig. 4(e), we observe that the network trained on ImageNet alone is not focusing on the whole bird and is only focusing on the body to make the decision; whereas the one trained with Anisotropic ImageNet is focusing on complete bird to make a decision.

We include more saliency maps on Sketch-ImageNet, and cases where ImageNet trained models are correct and our model fails in the supplementary material. We show more analysis about confidence of models and further analysis on transfer learning in the Supplementary material.

## 5 CONCLUSION

We propose to help a CNN focus on high level cues instead of relying on texture by augmenting the ImageNet dataset with images filtered with Anisotropic diffusion, in which texture information

is suppressed. Empirical results suggest that using the proposed data augmentation for pretraining self-supervised models and for training supervised models gives improvements across eight diverse datasets. Noticeably, the 11.4% improvement while testing the supervised model on Sketch ImageNet suggests that the network is indeed capturing more higher level representations as compared to the models trained on ImageNet alone.

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
