# OpenReview forum: "Learning Visual Representations for Transfer Learning by Suppressing Texture"
_ICLR.cc/2021/Conference — Reject_

### Official Review · AnonReviewer4 · 2020-10-27
**Simple yet effective idea, clearly presented.**

**Rating:** 5
**Confidence:** 3

**Review:**


In this paper, the authors propose suppressing texture (as low level cue) as it seems to provide shortcuts that prevent the network from learning higher level representations. The method achieves state-of-the-art on Sketch Imagenet (> +10%)

Pros:
- identifies the low level texture as a source of shortcuts which prevent learning, especially in unsupervised or self-supervised setting.
- simple idea, building on earlier research from the 90s; Gaussian blur was used before as source of augmentation; this work builds on top of the idea, and ensures that edges are preserved, mitigating the effect of texture influence.
- great improvement on Sketch ImageNet;

Cons / areas of improvement:
- the authors incorrectly claim testing on DTD dataset -- Please revisit Sec 4.2.1; Based on the citation, it looks like the authors are evaluating on a synthetic dataset, based on ShapeNet, with textures from DTD; Please clarify this in the paper.
- Table2: needs clarification, at least on DTD - is it using transfer learning on the right dataset?
- would be great to have one extra line with state-of-the-art for each of the datasets, e.g. in the form: 70.21 [citation]; when reporting numbers on DTD, it is fine to use only the first split (see SimCLR), as long as the evaluation protocol is specified in the dataset description paragraph.

Style, typos, glitches:
- wrong citation for DTD dataset; (Newel & Deng use objects from ShapeNet, and apply textures from DTD: https://www.robots.ox.ac.uk/~vgg/data/dtd/#citation).
- please explain all the notation in equations (1) and (2), e.g. \delta I, what is “c”, in equation (1) and (2); is it the same c?
Table1: citation or Stylized ImageNet; -- what is referred to as “Supervised”?

The idea seems simple, it is well validated through experiments, but needs some improvements in clarifying the evaluation on transfer to DTD dataset.

---

> ### Author Response · Authors · 2020-11-11
> **Fixed the citations of DTD dataset, added explanation on equations and state-of-the-art results. We fixed the definition of “supervised ImageNet”**
>
> We thank the reviewer for their thoughtful remarks and acknowledging our work on reducing texture as an effective idea. Below we address the raised concerns:
> 1) We have fixed the citation of the synthetic dataset which used  DTD for texture. We now refer to this dataset as synthetic-DTD dataset and have uploaded a new version of the paper which reflects this change.
> 2) The DTD dataset of Table 2 is using the original DTD dataset (https://www.robots.ox.ac.uk/~vgg/data/dtd/). We have followed the same convention as was done in SimCLR and reported the number on the first split in the DTD dataset. We have also added the state-of-the-art results from SimCLR in Table2 on all the datasets for which the authors had reported results.
> 3) Apologies for the confusion.
>    a)  Yes, c is the same in equations 1 and 2. It is known as the diffusion coefficient and it controls the rate of the diffusion in the image. We can see that the value of c will be low near the edges due to the high gradient. Equation-1 intuitively ensures that regions with a lower value of c will see less smoothing. Similarly, at points where we have less gradient, the smoothing is large and hence texture is reduced between the edges.
>    b) In the original paper, in equation 2 we wanted to make the dependence of c on the gradient explicit. We have updated the equation and have made the definition of c consistent. We have reflected these changes in the newly uploaded version of the paper.
> 4) By “supervised”, we meant ImageNet pretrained backbone and fine-tuned on VOC for object detection. We have replaced “supervised” by “supervised ImageNet” in Table 1 to clarify this.

---

### Official Review · AnonReviewer1 · 2020-10-27
**Interesting idea, however not enough technical contribution or experimental analysis.**

**Rating:** 4
**Confidence:** 4

**Review:**

The paper tackles the issue that CNNs may over-emphasize texture. The authors propose to suppress texture with anisotropic diffusion, as a way of data augmentation in training. The strategy is shown to be helpful for various downstream tasks, especially in a transfer learning setting.

I believe the paper has several advantages:
+ The proposed method is simple and well-motivated, and seems to provide consistent boosts across several settings.
+ The paper is clearly presented, figure 1 provides a good summary of the paper.
+ The visual analysis in sec 4.3 is quite interesting. The proposed technique is helpful in reducing the attention on texture in these examples.

Meanwhile, I have several concerns:
- technical novelty: the main contribution of the paper is proposing anisotropic diffusion as a way of data augmentation, which is somewhat trivial. Though it's quite simple, I feel it's quite heuristic and it's not supported by sufficient theoretical analysis or exploring of different design choices.
- experimental analysis: I am happy to see that the proposed technique achieved improvements across different settings. However, I think additional analysis is needed to better demonstrate that the representation focuses less on texture. Additional experimental analysis can also help to better understand the proposed technique and gain further insights.
- transfer learning: I am also not fully convinced by the effectiveness of the transfer learning setting. The experiments are conducted on Sketch-ImageNet, where images have little texture. Therefore, augmentation by suppressing texture naturally has an advantage in such a setting. I am a bit skeptical about the proposed technique in a more general setup, such as learning from synthetic data. (synthetic data usually has a different texture than real imagery). Besides, I think more baselines need to be added for transfer learning setting, such as techniques used in domain adaptation.

---

> ### Author Response · Authors · 2020-11-11
> **Novelty isn't always about complicated ideas! Clarification for design choices and analysis.**
>
> We thank the reviewer for their thoughtful remarks. Following are our comments for the concerns raised :
> *  **“technical novelty: the main contribution of the paper is proposing anisotropic diffusion as a way of data augmentation, which is somewhat trivial.”** ---We feel, along with AnonReviewer4, that the simplicity of our approach is a significant virtue.  Many widely adopted methods (eg., SimCLR[5]) are simple but elegant. On the contrary, simplicity is the strength of the proposed approach because it promotes easy and rapid adoption by the community. The proposed approach aims at balancing the trade-off between suppressing texture while maintaining the important high-frequency information “without a learning step” and Anisotropic diffusion optimizes for the same. Previous approaches such as Stylized ImageNet based augmentations totally breaks down for self-supervised setup as the deep CNN models can memorize the 79,434 textures and use them to solve the self-supervised tasks (Section 4.1 main paper). Similarly, gaussian blurring which produced state-of-art-results on SSL[5], doesn’t work well in supervised learning paradigms ( Table 4 main paper). Our approach works well in both cases because we strike the balance with the help of Anisotropic diffusion.
> * **“Though it's quite simple, I feel it's quite heuristic and it's not supported by sufficient theoretical analysis or exploring of different design choices.”:**
> We have shown the effects of different design choices for suppressing texture, in Table4, with the help of two different Anisotropic diffusion methods Perona Malik [1] and Michael Black [2]. Furthermore, we have also shown the effect of the number of diffusion iterations on the results in Table4. We have also shown comparisons with other classical texture suppression techniques such as bilateral filter and gaussian blur, and also the results of extreme bilateral+median filtering, referred to as cartoonized, in Table4. In all the comparisons, the proposed approach outperforms other approaches, which validates our choice for texture suppression against other similar approaches. \
> A detailed theoretical analysis would involve modelling the effects of such processing on large-scale real-world data-distributions and its, even more complex, interplay with the learning algorithms. While both these questions are extremely important, they are still wide open problems, and much more theory is needed to be developed before we can build concrete theoretical models for the same. Meanwhile, the research community is trying to understand these complex phenomena with the help of empirical analysis, which is evident from a series of impactful work for analyzing the role of texture information [7][9][10]. Such a line of work eventually leads the way towards more rigorous theoretical frameworks. Our work also falls in the same category of methods and once again learned reviewer’s concern about lack of theoretical analysis would deem all those impactful works unpublished. Lastly, our approach is based on Anisotropic diffusion principle, which is one of the better-understood frameworks for image processing and we hope that it inspires the community to take a deeper look into this direction to take us closer to a complete theoretical analysis. Therefore, we would request the reviewer to kindly re-consider their opinion around their concern for lack of theoretical analysis in light of the complexity of theoretical analysis and lack of such frameworks thereof.
> * **“additional analysis is needed to better demonstrate that the representation focuses less on texture.”**:
> In section 4.2.1, we have shown that our approach can reduce the reliance on texture information by experimenting with a recently introduced large-scale dataset, called Synthetic-DTD [3]. This dataset is created by taking 47 different textures from DTD dataset [4] and applying it to a 3D dataset of 10 classes, called ShapeNet [8] to yield the same object rendered with different view-points and multiple textures. Arguably, this dataset would pose extreme challenges to approaches that rely on texture specific information for classification. And, that’s precisely what we observe in the form of 10.41% improvement of our proposed approach over the ResNet-50 baseline.\
> We took motivation from this observation and carried additional experimentation on a popular dataset, named Geirhos Style-Transfer (GST) dataset [7], which comprises 1248 images of 16 classes from ImageNet dataset with shape and texture coming from different classes. This dataset is aimed at teasing apart the importance of shape vs texture for deep CNNs. We observe an improvement of 1.02% on identifying the class representing the shape, over a pre-trained ResNet-50 on normal ImageNet. This experiment shows that we can successfully reduce the reliance on texture.

---

> > ### Author Response · Authors · 2020-11-17
> > **We show consistent improvement for standard transfer-learning settings used popularly.**
> >
> > * **“I am also not fully convinced by the effectiveness of the transfer learning setting. The experiments are conducted on Sketch-ImageNet, where images have little texture. Therefore, augmentation by suppressing texture naturally has an advantage in such a setting.”** : The transfer-learning settings are adopted from the widely followed contemporary standard protocol put forward in SimCLR[5]. Our results under this standard transfer-learning setting demonstrate consistent improvements or ~1% across 5 different real-world datasets, shown in Table2. In addition to this standard-setting, we have also experimented with Sketch-ImageNet to showcase the improvements (11.41% over ResNet-50 baseline) offered by our approach in an extreme texture-less setting.
> >
> > * **I am a bit skeptical about the proposed technique in a more general setup, such as learning from synthetic data. (synthetic data usually has a different texture than real imagery)**: We have never claimed that our proposed work can solve the problem of learning from synthetic data, however, this investigation could be a useful direction for future-work. We thank the reviewer for pointing it out.
> >
> > * **“Besides, I think more baselines need to be added for transfer learning setting, such as techniques used in domain adaptation.”:** We have followed the most popular contemporary setting for transfer-learning in our work. While, in theory we see some value to reviewers proposal of adding domain-adaptation baselines, practically, it’s not possible to expect these many experiments in one work and we defer it for future work.
> >
> > [1]: Pietro Perona and Jitendra Malik. Scale-space and edge detection using anisotropic diffusion. IEEE Trans. Pattern Anal. Mach. Intell., 12:629–639, 1990. \
> > [2]: M. J. Black, G. Sapiro, D. H. Marimont and D. Heeger, "Robust anisotropic diffusion," in IEEE Transactions on Image Processing, vol. 7, no. 3, pp. 421-432, March 1998, doi: 10.1109/83.661192. \
> > [3]: How Useful is Self-Supervised Pretraining for Visual Tasks? arxiv - 2003.14323 \
> > [4] :M. Cimpoi, S. Maji, I. Kokkinos, S. Mohamed, , and A. Vedaldi. Describing textures in the wild. In Proceedings of the IEEE Conf. on Computer Vision and Pattern Recognition (CVPR), 2014. \
> > [5] Ting Chen, Simon Kornblith, Mohammad Norouzi, and Geoffrey Hinton. A simple framework for contrastive learning of visual representations, 2020a \
> > [6] Simon Kornblith, Jonathon Shlens, and Quoc V. Le. Do better imagenet models transfer better? \
> > [7] Robert Geirhos, Patricia Rubisch, Claudio Michaelis, Matthias Bethge, Felix A. Wichmann, and Wieland Brendel. Imagenet-trained cnns are biased towards texture; increasing shape bias improves accuracy and robustness. \
> > [8] ShapeNet: An Information-Rich 3D Model Repository. Angel X. Chang and Thomas Funkhouser and Leonidas Guibas and Pat Hanrahan and Qixing Huang and Zimo Li and Silvio Savarese and Manolis Savva and Shuran Song and Hao Su and Jianxiong Xiao and Li Yi and Fisher Yu \
> > [9] The Origins and Prevalence of Texture Bias in Convolutional Neural Networks: Katherine L. Hermann and Ting Chen and Simon Kornblith \
> > [10] Approximating CNNs with Bag-of-local-Features models works surprisingly well on ImageNet, Wieland Brendel and Matthias Bethge

---

> > > ### Author Response · Authors · 2020-11-20
> > > **Results on ImageNetC**
> > >
> > > Based on AnonReviewer1 concern we show additional experiments on dataset ImageNet-C[1], which evaluates model robustness to common corruptions. ImageNet-C[1] dataset has a total of 15 corruptions as shown in Table below. We calculate the error metric mCE in a manner similar to ImageNet-C[1]. We can see that by focussing less on texture our model is consistently more robust than the baseline Resnet model pre-trained on ImageNet data.
> > >
> > > |   | Noise    |       |         | Blur    |       |        |       | |    Weather   |       |        | Digital  |         |       |       |
> > > |-------|----------|-------|---------|---------|-------|--------|-------|---------|-------|-------|--------|----------|---------|-------|-------|
> > > |  Method| mCE(error)  | Gaussian | Shot  | Impulse | Defocus | Glass | Motion | Zoom  | Snow | Frost | Fog | Bright | Contrast | Elastic | Pixel | JPEG  |
> > > | Baseline(ImageNet) | 76.7 | 80 | 82  | 83 | 75 | 89 | 78 | 80 | 78   | 75    | 66  | 57  | 71  | 85 | 77  | 77    |
> > > | Anistropic(Ours)  |**74.85**| **72.70** | **76.65** | **78.27** | **73.73** | **87.97** | **75.99** | **79.70**| 80.33 | 78.02 | 68.05 | 58.30  | 71.34 | **82.64** | **70.88**| **68.16** |
> > >
> > > [1] Benchmarking Neural Network Robustness to Common Corruptions and Perturbations . Dan Hendrycks, Thomas Dietterich ICLR2019

---

### Official Review · AnonReviewer2 · 2020-10-28
**Interesting exploration of how texture suppression assists in domain generalization**

**Rating:** 7
**Confidence:** 5

**Review:**

##Updated Review##

I'd like to thank the authors for their response and am looking forward to seeing the result of their edge detection comparison.  I maintain my review of this paper.

# Main Idea
The main idea is that many networks learn shortcuts based on texture.  The authors propose to use anisotropic diffusion to augment training using images with suppressed texture.  This retains edge information without texture.  The authors claims to show that they achieve SOTA results on detection and classification in different datasets, and that their method is particularly effective for transfer learning tasks.

This work interpolates between work that tries to replace and resample texture and work that attempts to remove texture by focusing exclusively on edges with no other image content.  What is left after texture suppression is not just edges, but natural looking images with no discernible high frequency texture information (but high frequency edge information intact).  Color and low freqncy texture remains intact in these images.  In addition, shape information about the underlying natural objects is well preserved.  I think this is important because it has the effect suppressing specific high frequency texture while preserving low-frequency natural image statistics from within the training distribution.

Additionally, it should be noted that the authors trained their networks with a combination of standard image net and their augmented version. Which means that the original texture information is still present in at least a part of the dataset.  The training set now contains the same edge and shape information with frequency-localized texture variation.

# Interesting observations:
Interestingly, when comparing to the original Geirhos Stylized ImageNet performance on MoCo V2, the method presented here significantly outperforms.
### The authors note that:
One reason for this failure using the SIN dataset could be that the model is able to memorize the textures in the stylized images since it only has 79,434 styles. This is not a problem in the original fully-supervised setting where the authors used SIN for supervised image classification. In that case, the network can learn to ignore texture to discriminate between classes.

This points to the benefit of retaining some set of natural image statistics (from a larger distribution) within the training set.

Very intriguingly, the presented method also outperforms a network trained on SIN when evaluated on sketch-image net, a dataset comprised almost entirely of edge-like sketch images lacking almost all texture.  I find this result to be the most surprising of all those presented as these image lack all natural image texture statistics.  The performance is not out of this world, but it outperforms training on native image net and on SIN.  It would be interesting to see the comparison of performance here to a network trained exclusively on image-net images with only the edges extracted (perhaps using canny edge detectors).

# Weaknesses
Avoid sentences like the following as they are not completely supported by the work. "This forces the network to be less reliant on lower-level information to solve the pretext task and, hence, learn representations that focus on higher-level concepts.”

Sometimes the magnitude of differences in results between different methods (while being presented in tables as relatively modest improvements) are overstated in the text.  The results speak for themselves and there is no need to overstate small differences.

I find the saliency maps unconvincing (both because they are only a cherry picked subset) and because attending to the entire object does not mean the network is not also attending to the object’s texture.  Additionally, these methods are highly subjective in their evaluation (and likely to change depending on the method for saliency map production). Finally, the supplement shows examples that in fact don’t follow the established pattern presented in the original paper.  Finally, the analysis and results of the paper do not depend on this result and as such I think you should just remove it.

---

> ### Author Response · Authors · 2020-11-17
> **Will incorporate experiments using canny edge detector**
>
> We thank the reviewer for their thoughtful remarks and acknowledging our work on reducing texture as an effective idea and interesting exploration.
>
> We also thank the reviewer for the suggestion about training the network on images with only edges extracted using the Canny edge detector. We are working on this experiment and will include a comparison against this setting in the final version of the paper. Our hypothesis is that this network will perform better than our proposed approach on Sketch-Imagenet but its generalization might suffer on natural images.
>
> Finally, we appreciate the suggestions for stylistic changes in the paper. We will incorporate these and some other changes suggested by other reviewers in the final version before the end of the discussion period.

---

### Public Comment · ~Dinghuai_Zhang1 · 2020-11-11
**Related ICML work**

I enjoy reading your submission. I'm writing the comment to introduce our *HIGHLY* related ICML work:

Shi B. et al, Informative Dropout for Robust Representation Learning: A Shape-bias Perspective, ICML2020. https://arxiv.org/abs/2008.04254

Hope you'll enjoy reading our work and find it proper for reference.

---

> ### Author Response · Authors · 2020-11-11
> **Added citation for Informative Dropout work.**
>
> Thank you, Dinghuai, for your comment and for bringing this work to our attention. Indeed your work is similar to ours in a way that both methods are trying to reduce texture bias in the vision systems. But we follow classical texture reducing methods to reduce this bias while your paper uses dropout based methods which zeros out neurons by a high probability if the input patch contains less self-information.  We have included this reference in our paper and have updated our paper.

---

### Decision · Program_Chairs · 2021-01-07
**Final Decision**

**Decision:**

Reject

**Comment:**

This paper adopts an idea from 1990 for reducing reliance on texture, and shows that this idea improves the quality of visual representations in a variety of tasks. Initially reviewer scores were 7/5/4 but those improved slightly to 7/6/4 (changed in comment, not final review) after the rebuttal stage-- thus, one accept, one borderline, and one reject score. Reviewers have concerns about the great simplicity of the approach, where the only contribution is from prior work. Some reviewers request comparisons in a proper domain adaptation setting. While the large number of experimental settings somewhat balance out the concerns, overall, support for acceptance is not strong enough at this stage.